# Two Methods for Detecting PCM Residues in Vegetables Based on Paper-Based Sensors

**DOI:** 10.3390/s25082602

**Published:** 2025-04-20

**Authors:** Jiazheng Sun, Shiling Li, Xijun Shao, Mingxuan Fang, Heng Zhang, Zhiheng Zhu, Xia Sun

**Affiliations:** 1School of Agricultural Engineering and Food Science, Shandong University of Technology, Zibo 255049, China; 22503030059@stumail.sdut.edu.cn (J.S.); 22503030049@stumail.sdut.edu.cn (S.L.); 23503030366@stumail.sdut.edu.cn (X.S.); 2350303044@stumail.sdut.edu.cn (M.F.); 23403010317@stumail.sdut.edu.cn (H.Z.); 23403010287@stumail.sdut.edu.cn (Z.Z.); 2Shandong Provincial Engineering Research Center of Vegetable Safety and Quality Traceability, Zibo 255049, China; 3Zibo City Key Laboratory of Agricultural Product Safety Traceability, Zibo 255049, China

**Keywords:** immunochromatographic (IC) test, time-resolved fluorescence, AuNPs, procymidone, vegetables

## Abstract

Procymidone (PCM) is an effective, low-toxicity fungicide commonly used to control plant diseases in grains, vegetables, and fruits. Its usage has significantly increased in recent years, resulting in higher residues in vegetables. This study developed a sensitive and rapid immunoassay method utilizing a gold- and fluorescence-labeled monoclonal antibody (mAb) for detecting PCM residues in vegetable samples. Under optimal conditions, the fluorescent microsphere-labeled monoclonal antibody immunochromatographic strips achieved a limit of detection (LOD) of 1.67 ng/mL, with a visual LOD of 50 ng/mL. Intra-batch accuracy ranged from 94.98% to 103.82%, with a coefficient of variation (CV) of 1.97% to 8.26%. Inter-batch accuracy ranged from 96.16% to 102.51%, with a CV of 4.62% to 8.91%. The visual detection range of the gold nanoparticle-labeled monoclonal antibody immunochromatographic strips was 50 to 200 ng/g. The method demonstrated excellent performance in actual vegetable samples, confirming its applicability across various matrices. This dual-method approach enables rapid screening of negative samples with gold test strips, followed by accurate quantitative analysis of positive samples using fluorescent test strips, thereby enhancing efficiency and addressing diverse detection needs. Consequently, this method holds significant market potential for practical applications.

## 1. Introduction

Procymidone (PCM) is a low-toxicity organic fungicide used to control gray mold and other fungal diseases in vegetables [1,2]. Its widespread use has led to detectable residues in these products, raising potential health risks from accumulation in the human body. The high fat solubility of PCM makes it easy to accumulate in adipose tissue, and long-term low-dose exposure may lead to chronic toxic effects, including reproductive dysfunction, thyroid disease, and potential carcinogenic risk [2,3]. According to the International Codex Alimentarius, the amount allowed to be detected in vegetables is generally 0.2–5 mg/kg. This situation has increased the demand for effective monitoring of pesticide residues to ensure food safety. Developing a rapid and highly sensitive detection method for pesticide residues is essential. Current detection methods for PCM residues in vegetables include large-scale techniques like gas chromatography and liquid chromatography, as well as immunoassays such as enzyme-linked immunosorbent assays (ELISAs) and fluorescence immunoassays [4,5,6,7]. While recent advancements in sensor technologies—such as carbon nanotube-based electrochemical sensors for fungicides and optical sensors for pesticide detection—have demonstrated potential for rapid and sensitive analysis, methods still require specialized personnel and are unsuitable for on-site testing. Immunoassays, in contrast, offer advantages in terms of shorter detection times, portability, and cost-effectiveness [8,9]. However, they face limitations in quantification accuracy, shelf life, and specificity for certain sample types [10]. A comprehensive review of agricultural pesticide detection methods further underscores the need for integrated approaches that combine rapid screening with precise quantification [3].

The fluorescent microsphere-labeled monoclonal antibody immunochromatographic strip (FLMICS) is a biosensor based on antigen–antibody immunocompetition, characterized by a high sensitivity, short reaction time, simple operation, low cost, and real-time detection. It is widely used in food and medical industries [11,12,13]. The key components of FLMICSs are time-resolved fluorescence microspheres and monoclonal antibodies. In traditional methods, antibodies are adsorbed onto the microsphere surface through electrostatic and hydrophobic interactions, leading to high usage and low binding efficiency [14,15]. Europium oxide time-resolved fluorescence microspheres encapsulate trivalent europium compounds within polystyrene microspheres, with surface-modified carboxyl groups that couple with antibody amino groups to enhance binding stability [16,17]. The ability of antibodies to bind to antigens is crucial for test strip sensitivity. Mouse monoclonal antibodies are typically Y-shaped [18,19]; when the microsphere is coupled with the F_c_ region, the active sites on the F_ab_ region remain available for antigen binding (Figure 1A). Using goat anti-mouse IgG (GaMIgG) as a sandwich protein, which couples with time-resolved fluorescence microspheres while binding to the F_c_ region, maximizes the exposure of the antigen-binding domain, improving sensitivity [20,21,22].

The gold nanoparticle-labeled monoclonal antibody immunochromatographic strip (GLMICS) utilizes colloidal gold, where antigens or antibodies bind to gold nanoparticles to form conjugates that produce visible signals, allowing for detection through color changes [23,24,25]. Its advantages include rapid detection, simplicity, and low cost, making it ideal for immediate testing and rapid screening. Gold ions in HAuCl_4_ solution can be reduced to gold atoms by reducing agents [25,26], which aggregate into nanoparticles under electrostatic forces, forming a colloidal gold solution (Figure 1B). Particle size significantly affects optical properties and biocompatibility; both excessively large and small sizes can decrease detection sensitivity. Traditional methods for preparing gold-labeled antibodies face issues of uneven binding and poor storage stability [26]. Polyvinyl alcohol (PVA) is a polymer with good film-forming properties, commonly used in biomedical applications. PVA forms a protective layer on gold particles, ensuring dispersion [27]. Different concentrations and molecular weights of PVA can serve as stabilizers during preparation, ensuring uniform particle size and reducing aggregation [28]. The negatively charged surface of colloidal gold can form stable bonds with positively charged antibodies, enabling the preparation of gold-labeled antibodies [29].

This study employs GaMIgG as a sandwich protein and 1-(3-dimethylaminopropyl)-3-ethylcarbodiimide (EDC) as a crosslinking agent to couple time-resolved fluorescence microspheres with carbendazim monoclonal antibodies, creating an immunoprobe that enhances sensitivity and reduces antibody consumption. Citric acid serves as the reducing agent, and PVA as the stabilizer, to produce gold nanoparticles of appropriate size. After labeling the antibodies, colloidal gold immunochromatographic test strips are prepared, yielding superior detection results. The working conditions and preparation processes of the two types of test strips are optimized to establish standard curves and evaluate their performance. By testing various vegetable samples, the characteristics and advantages of colloidal gold and fluorescence immunochromatographic test strips are compared, demonstrating their applicability in detecting pesticide residues. According to the International Codex Alimentarius, the amount allowed to be detected in vegetables is generally 0.2–5 mg/kg. The combination of both test strips in food safety monitoring prioritizes colloidal gold test strips for rapid screening, followed by fluorescence test strips for precise measurement of contaminant levels, forming a multimodal detection method that leverages the advantages of both technologies. This model can achieve field detection of the target, where the LOD and detection width are enough to meet the minimum requirements of the detection limit of the target, and the cost per sample is reduced by 50–80% compared with other methods, for example Thermal Desorption–comprehensive gas chromatography–mass spectrometry (TD-cGC-MS), gas chromatography–mass spectrometry (GC-MS), Matrix Solid-Phase Dispersion–gas chromatography–Electron Capture Detection (MSPD-GC-ECD), enzyme-linked immunosorbent assay (ELISA), and high-performance liquid chromatography–Diode Array Detection (HPLC-DAD), while reducing detection time and costs (Table 1).

## 2. Materials and Methods

### 2.1. Materials and Instruments

PCM analytical standards, Triton, and bovine serum albumin (BSA) were acquired from Shanghai Aladdin Biochemical Technology Co., Ltd. (Shanghai, China). GaMIgG, procymidone complete antigen (PCM-BSA), secondary antibodies IgG, and procymidone monoclonal antibodies (PCM mAb) were obtained from Beijing Baidi Immunotechnology Co., Ltd. (Beijing, China). Time-resolved fluorescent microspheres and coating solutions were purchased from Shandong Lvdu Biotechnology Co., Ltd. (Shandong, China). Gold chloride (HAuCl4) was obtained from Sigma (St. Louis, MO, USA). 1-ethyl-3-(3-dimethylaminopropyl) carbodiimide (EDC, 98.5%) and N-hydroxysuccinimide (NHS, 98%) were sourced from Shanghai Macklin Biochemical Co., Ltd. (Shanghai, China). Other reagents, including sodium chloride, hydrochloric acid, sucrose, disodium hydrogen phosphate dodecahydrate, sodium hydroxide, and potassium chloride, were obtained from Sinopharm Chemical Reagent Co., Ltd. (Beijing, China). PVP-K30 was purchased from Tianjin Bodi Chemical Co., Ltd. (Tianjin, China). All solutions were prepared using ultrapure water (Millipore, Bedford, MA, USA).

The instruments included a self-cutting machine from Shanghai Jinsheng Biotechnology Co., Ltd., an XYZ 3D film coater and gold sprayer (HM3035), a UV-Vis spectrophotometer (RF-6000) from Shimadzu (Kyoto, Japan), a Sorvall ST16R centrifuge, and a transmission electron microscope from Thermo Fisher Scientific (Waltham, MA, USA). A three-purpose UV analyzer (ZF-1) from Jiangsu Qilin Su Meng Bell Laboratory Equipment Co., Ltd. (Jiangsu, China) was used for analyzing the FLMICS results. The additional equipment included a vortex mixer (KV37-Vortex Genie 2), a precision electronic balance (AL104) from Mettler-Toledo, a constant-temperature water bath (SHA-C) from Zhengzhou Honghua Instrument Co., Ltd., a high-temperature vacuum drying oven (DHG-9240A) from Shanghai Yiheng Technology Co., Ltd. (Zhengzhou, China), and an ultrasonic cleaner (KQ3200E) from Kunshan Ultrasonic Instruments Co., Ltd. (Kunshan, China).

### 2.2. Synthesis of Immune Probes

#### 2.2.1. Synthesis of FLM

A total of 1 mg of microspheres was washed twice using phosphate-buffered saline (PBS, 0.002 M) and resuspended in PBS. Next, 20 μL of NHS (20 mg/mL) was added, and the mixture was shaken for 1 min, followed by the addition of 5 μL of EDC (20 mg/mL) and rapid shaking for 8 min to activate the microspheres. After centrifugation at 12,000 rpm for 12 min, the microspheres were resuspended in 1 mL PBS and sonicated for 10 min to remove excess reagents. Subsequently, 4 μL of GaMIgG (5 mg/mL) was introduced and incubated at 37 °C for 2 h. To block excess binding sites, 100 μL of BSA (20 mg/mL) was added, and the mixture was incubated at room temperature for 1 h. The solution was centrifuged again, and the precipitate was resuspended in a stabilizing solution (0.5 g BSA, 0.1 g polyethylene glycol 400, 0.025 g casein, 0.05 g lysine, and 0.005 g HgS in 100 mL of 0.002 M PBS, pH 7.0) to achieve coupling of microspheres and GaMIgG. For the formation of FLM, 2 μL of mAb (2 mg/mL) was added, and the mixture was incubated at 37 °C for 2 h. To enhance stability, 2 μL of EDC (10 mg/mL) was introduced, and the mixture was incubated at 37 °C for another 2 h. Finally, 100 μL of BSA (20 mg/mL) was added, and the mixture was incubated at 37 °C for 2 h. The solution was centrifuged at 13,000 rpm for 12 min, and the precipitate was resuspended in the stabilizing solution and stored at 4 °C.

#### 2.2.2. Synthesis of GLM

A 100 mL solution of 0.01% HAuCl_4_ was boiled for 30 min on a magnetic stirrer. While boiling, 5 mL of 0.04 M citric acid trisodium solution and 0.5 g of PVA were added. The solution was boiled for another 15 min until it turned red, and then cooled to room temperature to yield a colloidal gold solution, stored at 4 °C. Generally, stable colloidal gold solution can be stored for 3–6 months. To 1 mL of this solution, 40 μL of 0.1 M K_2_CO_3_ and 2 μL of mAb (2 mg/mL) were added and shaken for 2 h. Next, 50 μL of BSA in boiling water was added to block excess binding sites, and the mixture was incubated at 37 °C for 2 h. The solution was centrifuged at 8000 rpm for 20 min, discarding the supernatant. Finally, 100 μL of gold-labeled solution (5% trehalose in borate buffer) was added to obtain AuNP-mAb, stored at 4 °C. The immunoassays were characterized using TEM and UV-Vis spectroscopy.

### 2.3. Assembly of Strips

#### 2.3.1. Assembly of FLMICS

Before assembly, the samples and conjugates were soaked in a sealing solution and dried at 37 °C for 2 h. The dried conjugates were then applied to the fluorescence probe and dried at 37 °C for 1 h. Using the XYZ 3D film coater and gold sprayer, PCM-BSA and GaMIgG were sprayed onto the test line (T line) and control line (C line) of the NC membrane at a rate of 0.6 μL/cm. The membrane was dried at 37 °C for 2 h. Because of its high-cost performance, chemical stability, and easy processing, Polyvinyl Chloride (PVC) has become the preferred material for test strip backers. The treated NC membrane, conjugate pad, and absorbent pad were sequentially adhered to the PVC backing plate (Figure 2). The adjacent parts were required to overlap by 0.1–0.2 cm. The strips were cut into 0.38 cm widths using a micro-cutting machine and stored in a dry, sealed bag at room temperature, protected from light.

#### 2.3.2. Assembly of GLMICS

Initially, goat anti-mouse secondary antibodies and coated antigens were individually sprayed onto the NC membrane to create the C and T lines, followed by drying at 37 °C for 2 h. A PVC backing plate was then prepared, and the sample pad, NC membrane, and absorbent pad were sequentially assembled and dried at 37 °C for 4 h or left overnight. Finally, the strips were cut into 0.3 cm widths using a micro-cutting machine and stored in self-sealing plastic bags.

### 2.4. Condition Optimization of Strips

To reduce sensor costs and enhance performance, the coupling pH for the PCM mAb was optimized. GaMIgG and time-resolved fluorescent microspheres were added to PBS at varying pH levels (5, 6, 7, 8, 9) to assess the relationship between pH and detection outcomes. We selected the scenario that produces the highest fluorescence intensity to determine the optimal coupling pH. The volume of GaMIgG used as the fluorescent probe significantly affected the sensitivity and detection range of the strips. After testing six gradient volumes, we selected the one with the highest T/C ratio as the optimal IgG volume. Once the optimal coupling pH and secondary antibody volume were determined, we minimized antibody usage while maintaining high sensitivity to achieve greater cost savings.

During the labeling of gold nanoparticles, a high pH can lead to excessive negative charges on both the antibody and gold nanoparticles, causing repulsion and hindering the synthesis of gold–antibody complexes. The pH was adjusted by varying the amount of K_2_CO_3_ (0.1 M) solution to achieve better coupling and color development. With the support of the results of the pre-experiment, the volume we chose was 5–10 μL. The coating concentration of the antigen on the T line was optimized to ensure adequate sensitivity. The concentration range of the antigen was determined to be 0.1–0.5 mg/mL. We selected the most uniform and appropriate color intensity strip of the test strip to determine the antigen concentration.

### 2.5. Performance Evaluation of Strips

The performance of the FLMICS was evaluated using a series of procymidone standards at concentrations of 0, 5, 10, 50, 100, 200, and 500 ng/mL. Each sample was tested in triplicate to calculate the average. The detection limit (LOD) was determined using the following formula:$$LOD=3\times SD\times 10^{\frac{z-b}{a}}$$
where *SD* is the standard deviation of the sample, *z* is the T/C ratio, *b* is the intercept of the standard curve equation, and *a* is the slope [11].

### 2.6. Detection of PCM in Leeks and Peppers

Negative samples of leeks and peppers were ground and mixed with anhydrous methanol. The mixture was sonicated and centrifuged to obtain the supernatant, which was filtered and diluted to pH 7.4 for analysis.

The supernatant was then loaded onto the assembled strips for the detection of PCM. The strips were incubated at room temperature for a specific period to allow for the binding of PCM to the immune probes. After incubation, the strips were washed thoroughly with phosphate-buffered saline (PBS) to remove any unbound substances. The presence of PCM was visualized using a fluorescence microscope or a spectrophotometer, depending on the type of immune probe used. The intensity of the fluorescence signal or the absorbance value obtained was used to quantify the amount of PCM present in the samples.

## 3. Results and Discussion

### 3.1. Principle Detection of Strips

In the FLMICS, fluorescent microspheres are conjugated with GaMIgG to create a fluorescent probe via mixing with PCM mAb. This conjugation ensures optimal exposure of the F_ab_ regions for specific antigen–antibody binding.

In the immunochromatographic test, both the T line and C line were chromogenic under the negative sample: the T line was chromogenic due to antigen–antibody binding, and the C line was chromogenic due to an the combination of unbound labeled antibody and a fixed antibody, verifying the normal function of the test strip. In positive samples, the target competes to suppress T-line color rendering, while C-line color is always displayed to ensure detection validity. The C line serves as the core quality control standard; if there is no color, the result is invalid.

Using a competitive principle, if the sample lacks the target analyte, the fluorescent probe binds to the immobilized PCM-BSA antigen on the test line (T line), producing a visible signal. However, when PCM is present, it occupies the binding sites on the fluorescent probe, preventing binding to the T-line antigen. As a result, the fluorescent signal intensity decreases with increasing PCM concentration, leading to a reduced T-line signal (Figure 3A,B).

The GLMICS operates on a similar competitive principle. When the test strip is immersed in the sample solution, the liquid migrates along the strip due to capillary action. The target analyte in the sample competes with the gold-labeled antibodies for binding to the immobilized antigens on the T line. This competition results in a color change on the T line, allowing for qualitative assessment of the presence of PCM in the sample (Figure 3C).

### 3.2. Characterization of Immune Probes

Transmission electron microscopy (TEM) images of fluorescent microspheres show the beads clustered together (Figure 4A). The fluorescent microspheres exhibit a maximum emission peak at 612 nm, with fluorescence intensities of 185,012 and 110,168 for the microspheres and the fluorescent probe, respectively (Figure 4B). Although the fluorescence intensity decreased after antibody conjugation, it remained sufficiently strong for effective detection.

TEM images of the gold nanoparticles (AuNPs) show a diameter of approximately 28 nm, which is consistent with the desired size for optimal sensitivity (Figure 4C). The uniformity and dispersion of the nanoparticles are crucial for maintaining their optical properties and biocompatibility. The UV-Vis spectroscopy results indicate a characteristic absorption peak at 520 nm (Figure 4D), confirming successful synthesis of the colloidal gold.

### 3.3. Optimization of the Working Conditions of Strips

The hydrophobicity and aggregation of the time-resolved fluorescent microspheres can be influenced by the pH of the phosphate-buffered saline (PBS) used during the coupling process. Various pH levels (5, 6, 7, 8, and 9) were tested to determine their effect on the fluorescence intensity of the assay. The optimal pH for coupling was found to be 8, which yielded the highest fluorescence signal (Figure 5A).

The volume of GaMIgG used as the fluorescent probe significantly impacts the sensitivity and detection range of the assay [35]. After testing various volumes, it was determined that 15 μL provided the best T/C ratio, maximizing the assay’s sensitivity (Figure 5B).

To enhance sensitivity while minimizing antibody usage, the optimal volume of PCM mAb was determined [36]. The best performance was achieved with 2 μL of antibody (Figure 5C,D), which provided a significant difference in T/C ratios between positive and negative samples.

During the labeling of gold nanoparticles, a high pH can lead to excessive negative charges on both the antibody and the nanoparticles, causing repulsion and hindering the formation of the gold–antibody complex [23]. The optimal K_2_CO_3_ (0.1 M) solution volume was 8 μL (Figure 6A), which resulted in the best colorimetric performance.

The concentration of the immobilized antigen on the T line was optimized to ensure adequate sensitivity [25]. The optimal concentration was found to be 0.3 mg/mL (Figure 6B), which provided a uniform and appropriate color intensity.

### 3.4. Performance Evaluation and Actual Sample Detection

The performance of the FLMICS was evaluated using procymidone standards at concentrations of 0, 5, 10, 50, 100, 200, and 500 ng/mL, with each sample tested in triplicate to calculate the average. The detection limit (LOD) was 1.67 ng/mL, with a visual LOD of 50 ng/mL. The assay showed a linear response from 5 ng/mL to 500 ng/mL, with a coefficient of determination (R^2^) of 0.99258 (Figure 7A,B).

The method was applied to actual vegetable samples, including leeks and peppers, with procymidone standards added at various concentrations. The results indicate a detection range of 200 to 1000 ng/g in leeks (Figure 7C) and 50 to 200 ng/g in peppers (Figure 7D). Accuracy was confirmed through recovery studies, with intra-batch and inter-batch coefficients of variation (CVs) ranging from 1.97% to 8.91% (Table 2).

## 4. Conclusions

In this study, we developed a rapid and simple method for detecting PCM residues in vegetables using a combination of GLM and FLM techniques. By introducing GaMIgG as a capture protein, we effectively oriented PCM mAb, reducing costs and enhancing sensitivity. The use of PVA as a stabilizer during colloidal gold preparation ensured uniform particle size and distribution. Compared with the reported immunochromatographic test strips (ELISA [33]), this study solved the problem of insufficient sensitivity or limited quantitative ability of traditional test strips through the collaborative optimization of time-resolved fluorescent microspheres and gold nanoparticles, and provided a scalable technical framework for synchronous detection of multiple pesticide residues.

The established competitive relationship between PCM concentration and T/C ratio demonstrates the method’s effectiveness. The LOD of the FLMICS was determined to be 1.67 ng/mL, with a visual LOD of 50 ng/mL. The visual detection range for the GLMICS was found to be 50–200 ng/g. The method exhibited excellent accuracy in actual vegetable samples, confirming its applicability across different matrices (Table 3). This dual-method approach allows for rapid screening of negative samples using GLMICSs, followed by accurate quantitative analysis of positive samples with FLMICSs, significantly improving efficiency and meeting diverse detection needs. Therefore, this method has broad market potential for practical applications. Moreover, the dual-method approach described here provides a comprehensive solution, addressing both the need for rapid screening and accurate quantitation. The integration of GLMICSs and FLMICSs not only enhances the overall detection capability but also offers flexibility in handling various sample types and concentrations. This versatility further solidifies the method’s position as a valuable tool in food safety and quality control, ensuring the reliability and accuracy of results across diverse matrices.

## Figures and Tables

**Figure 1 sensors-25-02602-f001:**
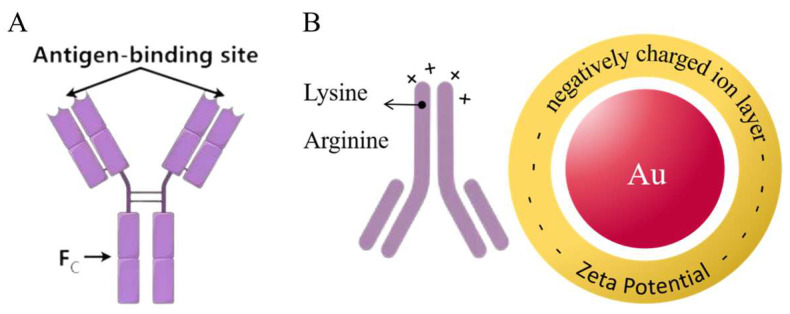
(**A**) Mouse monoclonal antibodies are typically Y-shaped, region F_ab_ has an antigen-binding site, and fully exposing this region can enhance the sensitivity of the immune probe. (**B**) The negative charges on the surface of colloidal gold particles and the positive charge regions on the surface of antibodies are attracted by Coulombic forces, forming an initial bond. Subsequently, hydrophobic interactions and van der Waals forces further stabilize the complex.

**Figure 2 sensors-25-02602-f002:**
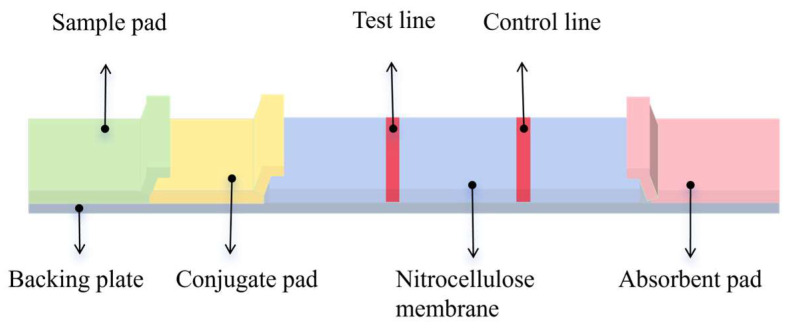
The structure of the test strip. The FLMICS uses dry detection with an added conjugate pad to better meet sensitivity requirements. The GLMICS uses wet detection to maximize cost and time savings.

**Figure 3 sensors-25-02602-f003:**
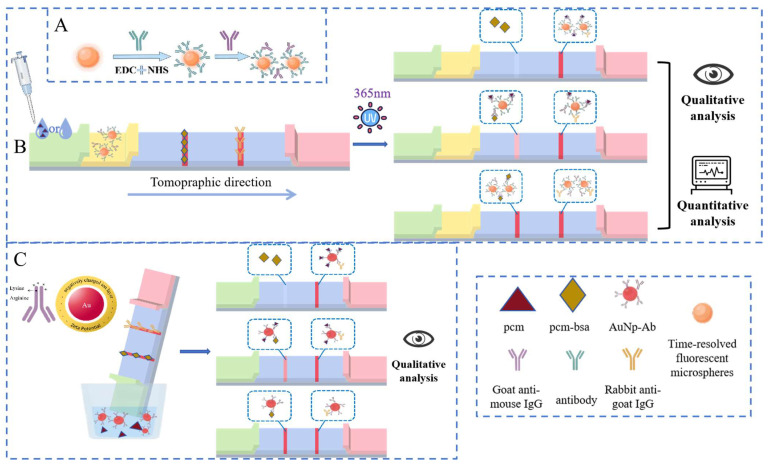
(**A**) Synthesis of the fluorescent probes by introducing GaMIgG as an intermediate, where the recognition site of the antigen is fully exposed. (**B**) The FLMICS based on the principle of immune competition enables qualitative and quantitative detection of target substances. (**C**) The GLMICS based on the principle of immune competition enables the qualitative detection of target substances.

**Figure 4 sensors-25-02602-f004:**
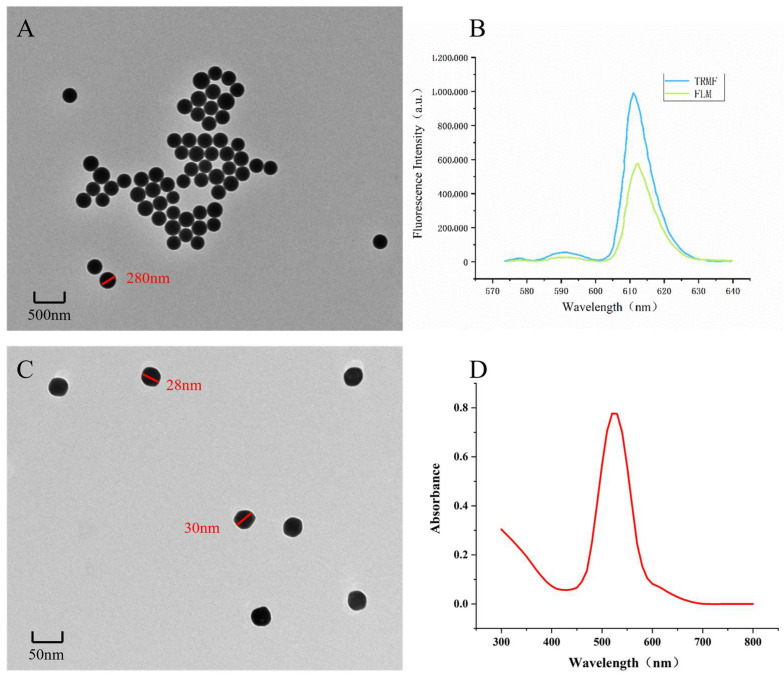
(**A**,**B**) The TEM image and fluorescence spectra of FLM; the preliminary results indicate that the microspheres have successfully coupled with mAb. (**C**,**D**) Through the TEM and UV absorption spectrum of GNPs, the colloidal gold particles can be seen to be uniform in diameter.

**Figure 5 sensors-25-02602-f005:**
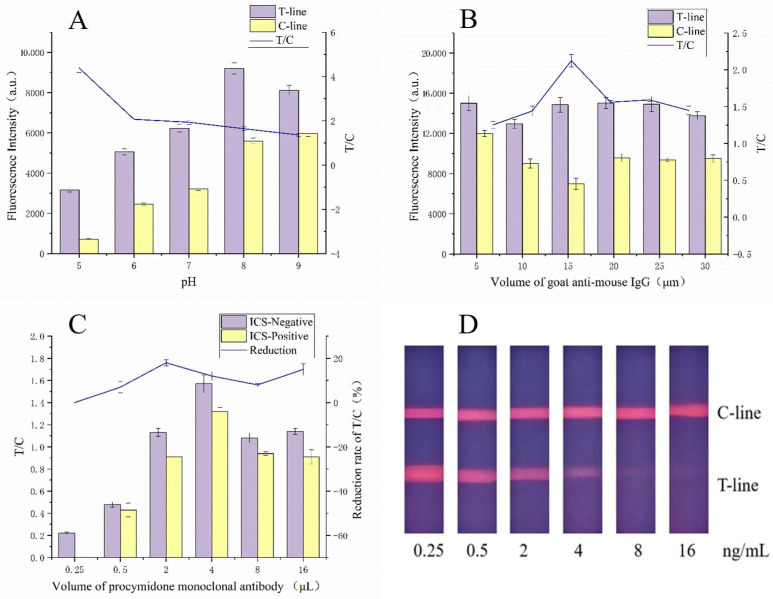
Parameter optimization of the FLMICS: (**A**) when the pH is 8, the fluorescence signal is the strongest and the T/C ratio is stable, so it is the best pH value; (**B**) when the volume of GaMIgG is 15 μL, the T/C ratio reaches its maximum; (**C**) when the volume of PCM mAb is 2 μL, the fluorescence intensity is excellent and the T/C ratio is stable; (**D**) the performance of the test strips.

**Figure 6 sensors-25-02602-f006:**
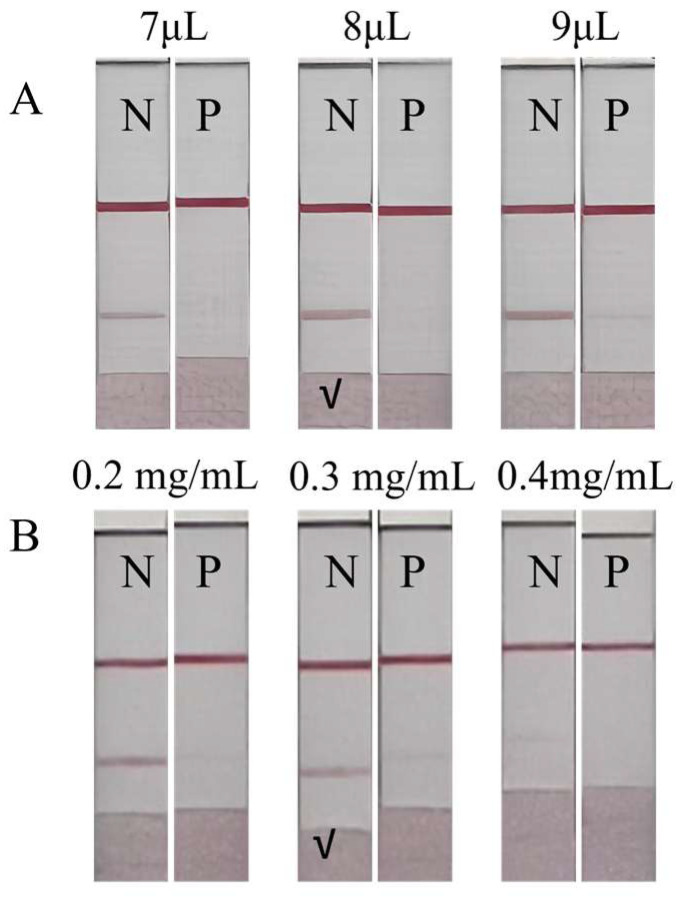
Parameter optimization of the GLMICS, N (negative control), and P (positive control). (**A**) The colorimetric effect is significant when the added volume is 8 μL. (**B**) The concentration of the antigen is selected as 0.3 mg/mL.

**Figure 7 sensors-25-02602-f007:**
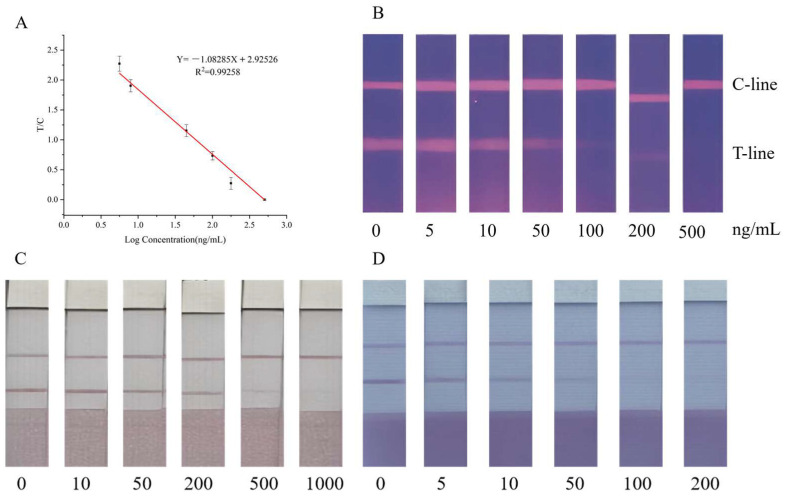
(**A**) The standard curve equation is Y= −1.08285 X + 2.92526. R^2^ is 0.99258; from this, the LOD of the FLMICS can be determined. (**B**) When the test strip shows 50 ng/mL, the brightness of the T line significantly decreases. (**C**) As the concentration of the standard increases, the color of the T line gradually becomes lighter. At a concentration of 200 ng/g, the T-line color begins to lighten significantly, and at a PCM concentration of 1000 ng/g, the T-line color can no longer be seen. Total disappearance indicates that the visual detection interval of the test strip of PCM’s leek sample strip is 200~1000 ng/g. (**D**) Similarly, the visual detection range for PCM’s pepper sample test strip is 50~200 ng/g.

**Table 1 sensors-25-02602-t001:** Comparison with other published methods.

Method	Application Scenario	Lod (ng/mL)	Detection Range (ng/mL)	Cost	Time	Reference
TD-cGC-MS	Laboratory testing	0.2	0.5~100	>USD 10 per sample	>10 min	[30]
GC-MS	Laboratory testing	0.44	1~100	>USD 10 per sample	>10 min	[31]
MSPD-GC-ECD	Laboratory testing	0.4	10~100	>USD 10 per sample	>10 min	[32]
ELISA	On-site testing	3	25~400	USD 8 per sample	10 min	[33]
HPLC-DAD	Laboratory testing	0.03	1.2~100	>USD 10 per sample	>10 min	[34]
FLM, GLM	On-site testing	1.67	5~250	USD 1 per sample	10 min	This study

**Table 2 sensors-25-02602-t002:** Stability experiment of TRFLIS.

Sample	Standard Addition Concentration (ng/mL)	Intra-Assay		Inter-Assay	
		Mean ± SD (ng/mL)	CV %	Mean ± SD (ng/mL)	CV %
leek	10	10.38 ± 0.58	5.58	9.85 ± 0.47	4.77
	50	50.77 ± 3.06	6.02	51.25 ± 2.37	4.62
	100	97.91 ± 3.42	3.49	96.16 ± 4.46	4.63
pepper	10	9.92 ± 0.82	8.26	9.90 ± 0.74	7.47
	50	51.20 ± 2.37	4.62	50.36 ± 4.06	8.06
	100	94.98 ± 1.88	1.97	97.18 ± 8.66	8.91

**Table 3 sensors-25-02602-t003:** Quantitative and qualitative test results of actual samples.

Sample	Standard Addition Concentration (ng/mL)	Qualitative	Quantitative	
			Detection Concentration (ng/mL)	Accuracy (%)
leek	0	−	0	100
	50	+	49.16	98.32
	100	+	97.03	97.03
pepper	0	−	0	100
	50	+	48.59	97.18
	100	+	103.26	103.26

## Data Availability

Data are contained within the article.

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
