# Peer review of "Two Methods for Detecting PCM Residues in Vegetables Based on Paper-Based Sensors"

_sensors, 2025, doi:10.3390/s25082602_

Round 1

Reviewer 1 Report

Comments and Suggestions for Authors

This study proposes a dual-method immunochromatographic approach based on GLMICS and FLMICS, combining the rapid screening capability of GLMICS with the precise quantification of FLMICS for the detection of procymidone residues in vegetables. The research demonstrates significant innovation and practical application value. However, the manuscript requires further revision due to several weaknesses in the methodology description, data analysis, and discussion sections, which need to be addressed to ensure its suitability for publication.

My detailed comments are as follows:

  1. The Abstract section contains too many abbreviations, which are not recognized terms. These abbreviations should be removed from the Abstract section and defined when they first appear in the text, except for the abstract.
  2. Some sentences were too long while some were extremely short. I'll advise the authors to engage a professional language editor to work on the manuscript, so that the language errors can be dealt with.
  3. The introduction of Table 1 lacks detailed explanation and comparison with current techniques, insufficient comparison with existing methods.It is recommended to provide a more comprehensive analysis to highlight the advantages of this study.
  4. While the study mentions the lower cost of single test strips, it does not systematically compare the time and economic savings with commonly used detection methods. It is suggested to include a detailed cost-benefit analysis to validate these claims.
  5. Some experimental steps are described too briefly, lacking critical details. It is recommended to provide more comprehensive descriptions to ensure reproducibility.
  6. The interpretation of Figure 5 is too brief. It is recommended to add more detailed explanations to improve clarity.
  7. The Table2 should be presented on a single page to avoid disrupting the flow of reading.
  8. Some sentences use inappropriate verb tenses, which obscures the research narrative. It is suggested to revise these for better clarity
  9. The Results and Discussion section needs to be expanded. The Results and Discussion should deals with the interpretation of the results in the light of previously published findings and must be clearly presented.
  10. Please add the Conclusions section. It must be fully supported by the results reported and should include the major conclusions, the limitations of the work and the future work.

Reviewer 2 Report

Comments and Suggestions for Authors

This study proposes a dual-method detection strategy for PCM residues in vegetables, combining colloidal gold immunochromatography and fluorescent immunochromatography. By introducing sandwich proteins and optimizing the preparation process of colloidal gold, the method fully leverages the advantages of both techniques, significantly enhancing detection efficiency and sensitivity. Through rapid screening with GLMICS followed by precise quantitative analysis using FLMICS, this approach effectively identifies pesticide residues in vegetables, addressing the market demand for efficient and cost-effective detection technologies. However, it is necessary to make a revision to the manuscript before it is considered for publication, see the following comments.

Q1. Although Table 1 includes a comparison with other detection methods, the introduction lacks a detailed description of the advantages of the proposed sensor. It is necessary to provide a comprehensive explanation of the sensor's strengths in the introduction.

Q2. In the optimization of test strip conditions, the optimization of the fluorescent method is presented in detail, while the optimization of the colloidal gold method lacks necessary explanations and a predefined range for optimization experiments.

Q3. The detection of PRM in leeks and peppers is described too briefly, lacking details on the target detection values and the interpretation of the results.

Q4. The caption of Figure 5 does not reflect the specific optimization details. Additionally, the description of the optimization approach in the main text is highly disorganized.

Q5. The Results and Discussion section lacks an explanation of the advantages of the detection method. It needs to be expanded to clearly and comprehensively demonstrate the application value of the sensor.

Q6. The Instruments and Materials section is disorganized in terms of language editing. Similar materials should be categorized, and attention should be paid to the use of verb tenses.

Comments on the Quality of English Language

The Introduction section contains sentences that are either overly verbose or excessively brief, leading to unclear expressions and a chaotic structure that affects the readability of the article. I recommend seeking professional English editing to address the language errors.

Reviewer 3 Report

Comments and Suggestions for Authors

  1. In "Keywords" replace the abbreviation "PCM" with the word "Procymidone".
  2. Are there maximum permissible concentrations of Procymidone in vegetables? How do the LOD and LR found in this study compare with maximum permissible concentrations?
  3. It is advisable to decipher the abbreviations in the "Method" column of Table 1.
  4. The text uses the abbreviation of the word "Procymidone" as both PCM and FRM (lines 39, 204, 261, 292-294). Make the abbreviation uniform.
  5. In section 2.2.2, indicate how long the gold sol retains its properties unchanged.
  6. "Negative samples of leeks and peppers were ground and mixed with anhydrous methanol" (line 205). Does methanol affect the T-line color intensity and sensitivity of Procymidone determination using the FLMICS and GLMICS.
  7. In section 3.1, describe not only the color of the T-line, but also the C-line with and without PCM on FLMICS and GLMICS.
  8. Why is the highest fluorescence signal obtained at pH 8 (lines 250-251)? Please, explain.
  9. The description of Fig. 5B and 5C is mixed up in the text. Replace Fig. 5B with 5C (lines 254-255) and also replace Fig. 5C with 5B (line 258).
  10. Decipher the designations N and P on the strips in the caption to Fig. 6.
  11. Has the selectivity of the developed Paper-Based Sensors been tested against structural analogues of Procymidone, such as Vinclozolin, Chlozolinate, Iprodione, Imidacloprid?
  12. The fluorescence intensity of the T-line and C-line is a dynamic process that changes over time. How was the moment of fluorescence intensity measurement determined?
  13. What is the lifetime of the developed Paper-Based Sensors?
  14. Section 4 should be «Conclusion», not “Discussion”.

Reviewer 4 Report

Comments and Suggestions for Authors

The presented article is devoted to an urgent problem - the assessment of the fungicide procymidone in plant products, as it can be potentially dangerous when it accumulates in the body and due to its wide usage, it becomes a threat to human health. The article is well structured, the material is presented consistently, and the main conclusions correspond to the results of the study.

However, for a clearer understanding of the conducted research and its practical application, it is advisable to make minor adjustments to the article.

1) The article provides a fairly detailed description of classical methods for determining pesticides and fungicides in plants, but for a full understanding of the development of analytical methods for this class of substances, it would also be useful to compare the methods with other types of sensors in Table 1, for example

  1. Zamora-Sequeira, R., Starbird-Pérez, R., Rojas-Carillo, O., & Vargas-Villalobos, S. (2019). What are the main sensor methods for quantifying pesticides in agricultural activities? A review.Molecules24(14), 2659.
  2. Ilager, D., Malode, S. J., & Shetti, N. P. (2022). Development of 2D graphene oxide sheets-based voltammetric sensor for electrochemical sensing of fungicide, carbendazim. Chemosphere,303, 134919.
  3. Chethan, B., Sunilkumar, A., Prasad, V., & Thomas, S. (2024). Surfactant electrochemical sensor based on carbon nanotubes for the analysis of fungicides. In Advances in surfactant biosensor and sensor technologies(pp. 217-228). Cham: Springer Nature Switzerland.

This would also be useful to justify the conclusion about the potential for using the proposed method in the market.

2) The discussion mainly discusses the results of the study, and it is not clear how this relates to previously conducted studies on the topic, with the exception of three papers. Why, in the authors' opinion, has the proposed approach not been used earlier? To make it clear what exactly the authors' contribution to expanding knowledge on the topic of fungicide and pesticide analysis is. What concentration of procymidone is potentially dangerous for humans and in what time can it be achieved? What restrictions are imposed on the fungicide content in plants in this regard? And how will the proposed method allow this to be avoided?

3) For better understanding by a wider class of specialists, it is better to provide a separate explanation of the abbreviation of all methods in Table 1 as the note.

4) Instead of a Сonclusion, section 4 named as Discussion, apparently a typo.

Reviewer 5 Report

Comments and Suggestions for Authors

The work developed a new device for detecting a fungicide, which if accumulated in the body can be an endocrine disruptor. Therefore, some few lines more to highlight the relevance of this analyte target should be provided. Also, the table 1  should be presented in discussion section.

I dont like the title, the dual concept was not clear for me. Moreover, since the device must be prior constructed, the final words "..and conctruction of paper based sensors" can be removed.

The keywords should not contain abbreviations.

In lines 39, 204, 261, 292, 293: PRM must be PCM?

In lines 151, 265: K2CO3 

In line 171: PVC, as well as, Pol-72 yvinyl alcohol (PVA), is not so well known. Maybe, besides the description, it could be put in figure2.  PVC backing plate.

Round 2

Reviewer 4 Report

Comments and Suggestions for Authors

The changes made allowed to clarify many issues and improve understanding and presentation of the research methodology. The conducted research and the obtained results undoubtedly have scientific and practical significance for the development of new methods for rapid analysis of the presence and determination of the amount of pesticides and fungicides.

However, the authors mentioned in their response that they compared their developments with developments based on other types of sensors and included their characteristics in Table 1, although these additions to Table 1 were not found in the text of the amended manuscript.

Also, adding recommended works to the list of references without changing the content of the text when citing seems not entirely correct.
Elimination of these minor comments will fully clarify the significance of the publication.
